# Estimating COVID-19 vaccine uptake and its drivers among migrants, homeless and precariously housed people in France

Thomas Roederer [1✉], Bastien Mollo [1,2,3], Charline Vincent [1], Ghislain Leduc[1], Jessica Sayyad-Hilario[1], Marine Mosnier[4] & Stéphanie Vandentorren[5,6]

## Abstract

**Background** Migrants, people experiencing homelessness (PEH), or precariously housed (PH) are at high risk for COVID-19 infection, hospitalization, and death from COVID-19. However, while data on COVID-19 vaccine uptake in these populations are available in the USA, Canada, and Denmark, we are lacking, to the best of our knowledge, data from France. **Methods** In late 2021, we carried out a cross-sectional survey to determine COVID-19 vaccine coverage in PEH/PH residing in Ile-de-France and Marseille, France, and to explore its drivers. Participants aged over 18 years were interviewed face-to-face where they slept the previous night, in their preferred language, and then stratified for analysis into three housing groups (Streets, Accommodated, and Precariously Housed). Standardized vaccination rates were computed and compared to the French population. Multilevel univariate and multivariable logistic regression models were built. **Results** We find that 76.2% (95% confidence interval [CI] 74.3–78.1) of the 3690 participants received at least one COVID-19 vaccine dose while 91.1% of the French population did so. Vaccine uptake varies by stratum, with the highest uptake (85.6%; reference) in PH, followed by Accommodated (75.4%; adjusted odds-ratio = 0.79; 95% CI 0.51–1.09 vs. PH) and lowest in Streets (42.0%; AOR = 0.38; 95%CI 0.25–0.57 vs. PH). Use for vaccine certificate, age, socioeconomic factors, and vaccine hesitancy is associated with vaccination coverage. **Conclusions** In France, PEH/PH, and especially the most excluded, are less likely than the general population to receive COVID-19 vaccines. While vaccine mandate has proved an effective strategy, targeted outreach, on-site vaccinations, and sensitization activities are strategies enhancing vaccine uptake that can easily be replicated in future campaigns and other settings.

**Plain language summary**

Vulnerable populations, such as people experiencing homelessness, are less likely to have a COVID-19 vaccine. We aimed to identify potential reasons for this, by interviewing homeless/precariously housed people in France. We found that although most homeless people have been vaccinated, vaccination rates are lower than the general population. Among the homeless, the least likely to be vaccinated are those living on the streets. The need for vaccine certificates and the support of social workers are positive drivers of vaccine uptake, while influence from family/friends, vaccine hesitancy and fear of the vaccine negatively affect uptake. Providing vaccines on-site and tailoring programs to better target these vulnerable groups should be priorities. Raising awareness by involving trusted third parties is also key to countering negative vaccine beliefs. Our insights apply beyond the COVID-19 crisis, when routinely supporting the health of vulnerable populations.

[1] Epicentre, Paris, France. [2] Médecins Sans Frontières, Paris, France. [3] Infectious and Tropical Diseases Department, Bichat-Claude Bernard Hospital, AP-HP, Paris, France. [4] Prospective et Coopération, Marseille, France. [5] Santé Publique France, Saint Maurice, France. [6] University of Bordeaux, INSERM UMR 1219-Bordeaux Population Health, Bordeaux, France. ✉email: thomas.roederer@epicentre.msf.org

Evidence from the early COVID-19 waves suggests that population subgroups, such as people experiencing homelessness (PEH) or precariously housed (PH), are disproportionately exposed to infection[1–3] and the severe forms of the disease[4–8], as well as suffering from greater mental health and social impacts[7,8]. Transmission risk is worsened by factors specific to these groups, such as precarious living conditions, high population density, need to access food distribution services, poor access to sanitation and hygiene, and difficulties accessing care[1–3,5,6]. COVID-19 prevention measures such as social distancing and self-isolation are challenging to maintain for such groups[1,3,5,6].

In 2021, highly efficacious COVID-19 vaccines became available, providing strong protection against severe disease, hospitalization, and death. It is already known that PEH/PH tend to uptake vaccination against diseases other than COVID-19 to a lower degree than the general population[9–12]. Obstacles to vaccination for PEH/PH include practical barriers and service limitations[9,11,12], suboptimal experiences with vaccines or health services[9–11], and modern medicine/vaccine hesitancy[9,10,12]. Structural obstacles include poor housing[9,12,13], inadequate medical coverage and access to care[9–14], and not considering disease prevention a priority[12,14]. Migrants and refugees also encounter obstacles to vaccination within host countries, including language barriers and lack of access to information[9,10,12–15], not considering health as a priority[15], high mobility/no fixed address[10,11,13–15], and lack of suitable healthcare providers[13–15]. Moreover, migrants and refugees may also be reluctant to take up vaccination, for fear of deportation while waiting for the right to reside[14,15].

France's COVID-19 vaccination strategy was implemented in five stages from January 2021 onwards, with vaccination cost-free, regardless of medical coverage or administrative status[16] (detailed timeline in Supplementary Notes 1). Expansion to all adults aged over 18, including PEH/PH, was then implemented in June 2021. Other important developments were the introduction of the Pass Sanitaire (vaccine certificate) in July 2021 and the introduction of third (booster) doses for all adults, including PEH/PH, from November 2021 on.

Residents of migrant workers' hostels and homeless people >55 yo were originally supposed to be prioritized during the first round (January–April 2021). However, the lack of vaccines and of actors able to perform vaccination on the field nullified the prioritization. Médecins Sans Frontières and other Non-Governmental Organizations (NGOs) officially started vaccinating migrants and homeless people in late May[16]. Strategies towards PEH/PH included the management of vaccination centres dedicated to PEH/PH, sensitization, mobile vaccination teams, and physical accompaniment to vaccination centres[16] (Supplementary Note 2).

The PEH/PH population was estimated to be around 250,000 people in France in 2021, with ~150,000 in the Ile-de-France region and Marseille. Of these, 25,000 are housed in workers' hostels, 50,000 in centres for asylum seekers and emergency shelters, and 35,000 in social hostels[17]. 2800 are estimated to be permanently living rough in Paris, and around 1500 in Marseille[18]. Few global data are available on vaccine uptake among PEH/PH; we identified only six quantitative[19–24] studies including a nationwide retrospective study in Denmark and a large retrospective survey in Canada, that all highlighted a markedly lower vaccine uptake compared to general populations, without reports on associated factors or barriers to COVID-19 vaccination. Unfortunately, no official French data for COVID-19 vaccine coverage in migrants, homeless, or roofless populations exist. Moreover, data from European countries with similar migration and homelessness profiles (eg. Belgium, Germany, England) are so far absent as well.

Given the higher risk of COVID-19-related morbidity and mortality and the lack of understanding of COVID-19 vaccination drivers, we aimed to estimate vaccination coverage in PEH/PH in Paris and Marseille areas and to analyse factors associated with vaccination status.

Our study highlights that PEH/PH in France is less likely to receive COVID-19 vaccination than the general population and that people living in the streets, in camps, or in squats are the least likely to be vaccinated. Taking housing into consideration plays a major role in vaccination campaigns.

Multivariate analysis shows that external factors like the need for a vaccine certificate, medical coverage, medical and/or social support, and on-site vaccinations are all associated with vaccine uptake, while individual factors such as age, opinion on vaccination, and fear of COVID-19 vaccines are clear drivers.

## Methods

**Survey design and sampling strategy**. We performed a cross-sectional study between 15 November and 22 December 2021 in the Ile-de-France region and the city of Marseille.

Inclusions were performed at the place participants last slept the night. Inclusion criteria were to be aged >18 and in full capacity to give consent. Exclusion criteria were the participant's inability to answer (due to obvious mental disorders, drug or alcohol influence) or impossibility to conduct the interview for safety reasons (to the interviewer's discretion) or language barriers. Interviewers were trained to ensure each participant could only be included once.

Recruitment sites were defined and stratified using the European Typology on Homelessness and Housing Exclusion (ETHOS) typology (Fig. S1 in Supplementary Information). Four strata in Ile-de-France included (a) migrant worker hostels; (b) emergency shelters and centres for asylum seekers; (c) social hostels and similar facilities; and (d) individuals permanently sleeping rough (in the street, in parks, in the subway), living in informal camps, squats or slums. Another stratum was a subsample of a cohort following homeless and migrant populations living rough, in squats, emergency shelters, and transitional housing in Marseille.

Migrant Workers' Hostels are a France-specific type of facility built by the French authorities during the 1950s, which purpose originally was to offer a short-term housing solution for a supposedly temporary migrant labour force, usually from North and West Africa. MWHs are often managed by state agencies, and sometimes private organizations. Residents are paying rent for very small and often crowded rooms (Supplementary Notes 3).

We estimated sample size per stratum, based on assumptions reflecting vaccine hesitancy reported in the literature, and considering 80% power, design effects of 3% and 5% accuracy. Total estimated required sample size was 3751 (Supplementary Methods 1).

In Ile-de-France, we built sampling frames for each stratum using data provided by various actors involved with these populations; these listed locations, and the size of each site (Supplementary Methods 1). In each stratum, recruitment sites were randomly selected (first stage) proportionally to their size.

Sample size per site was calculated in proportion to the expected site population, with participant sampling in the second stage depending on site type. In shelters, migrant workers' hostels, and centres for asylum seekers, individuals were selected using simple random sampling when resident lists or room lists existed, and systematic random sampling otherwise. To ensure the selected person was included, sites were visited repeatedly at different times, including weekends and evenings. If individuals

were absent or declined to consent, selected individuals were replaced by another sharing that room or the one adjacent.

For people living in the streets/camps/slums, we obtained an exhaustive census map recording all homeless and migrants living in subdivisions of Paris in March 2021[18]. All individuals were systematically invited to participate (exhaustive sampling) until the stratum sample size was reached. In cases of refusal, the next person apparent was interviewed.

In Marseille, a local NGO record all PEH/PH and migrants living in the urban area to provide healthcare and legal advice. This NGO was the operational partner for the study. We drew a subsample from their cohort, using simple random sampling, with planned replacements for refusal/absence.

**Outcomes and definitions**. Vaccine uptake was the main outcome, defined as the uptake of at least one COVID-19 vaccine dose, irrespective of brand or type. Vaccine coverage was defined as a full schedule of COVID-19 vaccine, i.e. at least two doses of messenger RNA vaccine (usually Pfizer), one dose of Janssen vaccine, or one dose of any vaccine following prior to COVID-19 infection.

**Data collection**. After obtaining verbal informed consent, questionnaires were administered by trained interviewers in the participant's preferred language. Interviews were conducted in French, English, Arabic, Farsi, Spanish, Turkish, Wolof, and Pulaar or were translated by phone into any other language. Responses were recorded using tablets (interview form in Supplementary Data 1).

COVID-19 vaccination status was verified via the national vaccine certificate ("Pass Sanitaire"), either on the "TousAnti-Covid" phone app or the paper version of the certificate."Questionnaire topics covered sociodemographics (age, gender, administrative status, native language, duration in France), housing (type of residence in the past 3 months, mobility), participants' views about vaccines (general and COVID-19-specific), vaccination (status, place, date, reasons for vaccination and non-vaccination), health-related information (history of COVID-19 infection and/or hospitalization, medical coverage), sources of COVID-19 vaccination information (internet, TV/radio, relatives, etc.), finances and related (work, source of income, source of meals), support and coping mechanisms (food distribution, support organizations), moral and material support from relatives or social workers, health literacy and discrimination. Questions were selected following discussions with a panel of experts in social determinants of homeless and migrant people's health, using the framework conceptualized by the World Health Organization[25].

We also collected information on recruitment sites, covering the distance to vaccination sites, including those for the general population as well as those dedicated to PEH/PH.

**Grouping for analysis**. Sampled populations are mobile, with individuals often staying at a site for just a few days. For analysis, we recombined strata into three categories, based on the most reported type of residence over the last 3 months, irrespective of recruitment location. The three groups comprised Precariously Housed (individuals renting their own accommodation or housed in a migrant workers' hostel); Accommodated (temporarily hosted in asylum seekers' centres, emergency shelters, or social hostels); and Streets (individuals sleeping rough, in camps, in squats or in slums).

**Statistics and reproducibility**. For variables with a high number of missing values (>5%), the missing values mechanism was assumed to be the MAR mechanism (missing at random), which was verified. For each variable, several imputations methods were compared (multiple regression, random forest) and the one which gave the lowest error rate was retained. Only imputed variables with an error rate lower than 20% were used for the multivariable model.

Summary measures for main outcomes (vaccine uptake and coverage) were calculated by location, stratum, and participant characteristics, and expressed as estimates with Clopper–Pearson 95% confidence intervals (CI). Descriptive analyses were performed taking into account sampling weights for inclusion probability, and clusters for variance estimation. Exact Fischer's Test was performed when the number of observations was insufficient and Pearson Chi-Squared Tests otherwise for univariate and multivariable modelling, unweighted analyses were performed using Likelihood Ratio Tests to qualify potential associations.

Sample vaccine uptake and coverage were compared to the French general population. Weighted direct standardization by age category was performed, using age cut-off: 18, 25, 40, 55, 65, >65; estimates and 95% CI were computed for the overall study population and for each stratum (Supplementary Methods 2).

Univariate logistic regression analysis explored vaccine uptake-associated factors for all strata combined. A multilevel multivariable logistic regression model was constructed with random intercepts for specific recruitment sites to account for clustering and random effects on several variables after testing for validity (see below). We included variables that could explain differences in vaccine uptake proportions.

Only variables with $p \leq 0.2$ after univariate were retained in the full multivariate model. Multicollinearity was verified before model selection and variables with a variance inflation factor (VIF, for continuous variables) >5 or a generalized VIF (GVIF, for discrete variables) >2.5 were dropped. Random effects were tested on the full model and selected to minimize the second-order Akaike Information Criteria (AICc). After random effects selection, fixed effects were selected with a backward procedure, minimizing AIC.

Some potential confounding factors or factors known to be linked with vaccine uptake in the literature (e.g. age, sex, health literacy, or social support) that were not significant after univariate analysis were kept for model adjustment. Moreover, personal opinions and distances to vaccination sites variables appeared too important/interesting to not include.

Validation of the final model consisted in analysing standardized residuals (overdispersion, distribution, outliers) and verifying coefficients of determination. Further details on adjustment for confusion and multicollinearity are available in Supplementary Methods 2.

Additional analyses included univariate/multivariate stratified analyses following the same plan, negative binomial regressions on the total number of vaccinated individuals by site as a count outcome, and site-related variables as covariables (Supplementary Methods 2). Data were analysed using Stata v.16 software (StataCorp. 2019. College Station, TX) and R (v3.6.2).

A reproducibility guide for figures and tables is available in Supplementary Methods 3.

**Ethics**. The study protocol was approved by the Comité de Protection des Personnes III, Ile de France, Paris on 13 August 2021 (ref. 2021-A01960-41). Participants were not interviewed before giving oral informed consent.

**Reporting summary**. Further information on research design is available in the Nature Portfolio Reporting Summary linked to this article.

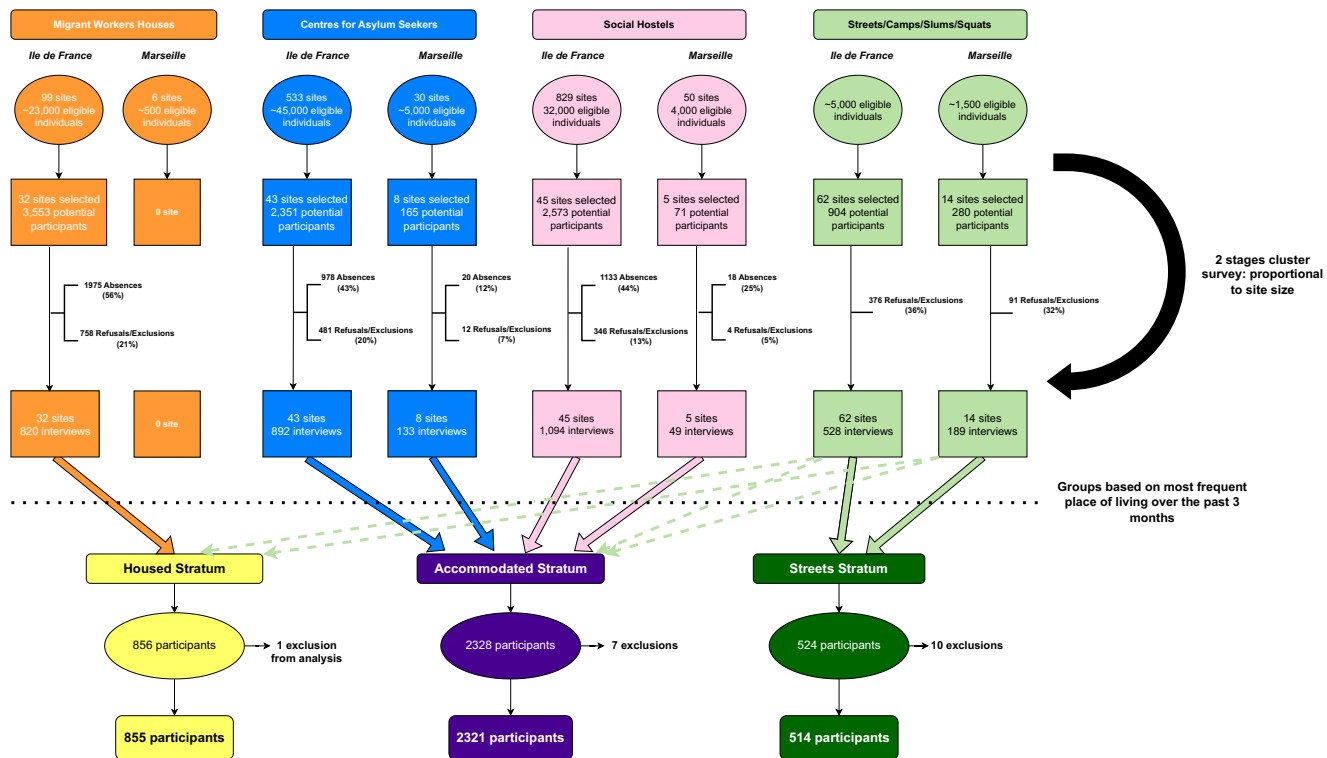

**Fig. 1 Study flow chart.** Orange, Blue, Pink and Light Green items represent the original strata prior to statistical analysis. Yellow, Purple and Dark Green items represent groups created for the analysis, based on the most frequent places of living over the past 3 months.

## Results

**Study population**. 9897 individuals were initially selected, of whom 4124 (42%) were absent, 1757 (17.7%) refused to participate and 326 (3.4%) were subsequently excluded.

A total of 3690 individuals were surveyed and kept for analysis: 3319 in Ile-de-France in 182 centres for asylum seekers, social hostels, migrant workers hostels, and recruitment locations in the streets, camps, or slums (representing 3.2% of the total eligible population and 9.1% of eligible sites) and 371 in Marseille in 27 sites (3.4% of eligible population and 21% of eligible sites) (Fig. 1 for study flow chart and Supplementary Figs. S2 and S3 for maps).

**Characteristics of study participants**. Of the 3690 surveyed individuals, 855 comprised the Precariously Housed stratum, 2321 the Accommodated stratum, and 514 the Streets stratum.

53.7% of study participants were male with very few women included in the Precariously Housed stratum (4.6%). The weighted mean age was 41 years (95% CI: 39.9–41.8). Precariously Housed participants were older than those in other strata (22.5% >65 y vs. 4.4% in Accommodated and 3.5% in Streets respectively; $p < 0.001$). In the Precariously Housed stratum, individuals commonly originated from West Africa (48.0%) or Central/Southern Africa (26.1%); 61.1% had been in France for >10 years, and 62.5% had official documentation. In Accommodated stratum, geographic origins were similar (West Africa 43.1%; Central/Southern Africa, 20.1%), but 62.3% had been in France for <10 years (and 38.0% were undocumented, with 31.2% seeking asylum. 48.2% of the Streets stratum were French or EU citizens and this stratum had the highest proportion of recently arrived migrants (27.7%).

Almost all Precariously Housed participants (92.8%) and the majority of Accommodated participants (63.5%) were able to buy their own meals, while 52% of Streets participants could, and 43% of them resorted to panhandling.

While most participants in the Precariously Housed stratum (67.4%) and half of the Streets participants (46.5%) reported living alone, most Accommodated participants shared a room with others (78.9%). Supplementary Data 2 summarizes previous results while other variables are described by strata in Supplementary Data 3.

**Vaccination**. 76.2% (95% CI 74.3–78.1) of surveyed individuals reported receiving at least one COVID-19 vaccine dose in 2021; 73.1% (95% CI 69.6–76.5) reported receiving a full vaccine schedule. Vaccination status was verified via a certificate (paper version or electronic) for the majority (82.2%; 95% CI 79.8–84.6).

Vaccine uptake varied significantly by stratum (chi² $p < 0.001$) with an overall design-effect (DE) of 1.85 (Supplementary Data 4). Uptake was highest (85.6%; 95% CI 83.0–88.2; DE: 2.78) among Precariously Housed individuals, followed by Accommodated (75.4%; 95% CI 73.0–77.8; DE: 2.51) and lowest in Streets participants (42.0%; 95% CI 34.3–49.7; DE: 1.08). Variability of vaccine uptake by recruitment site was high, especially in the Streets stratum (Supplementary Fig. S6).

Standardization of vaccination rates allowed for comparison with the French population: overall, vaccine uptake in PEH/PH was 79.9% (95% CI: 79.2–80.6), a proportion significantly lower than uptake in the general population (91.1%). Overall, for each stratum and for each age category, vaccine uptake was delayed by roughly two months as compared to the general population (Figs. 2, 3).

Most participants received their injections after the vaccine certificate was first announced on July 12 (23.7%; 95% CI: 21.5–26.0) or extended on August 9 (44.2; 95% CI: 41.2–47.2). Vaccination sites for surveyed individuals included vaccination centres open to all (55.1%; 95% CI: 52.0–58.2), followed by healthcare services available to those with medical coverage (drugstores, general practitioners—GPs, hospitals; 23.9%; 95% CI

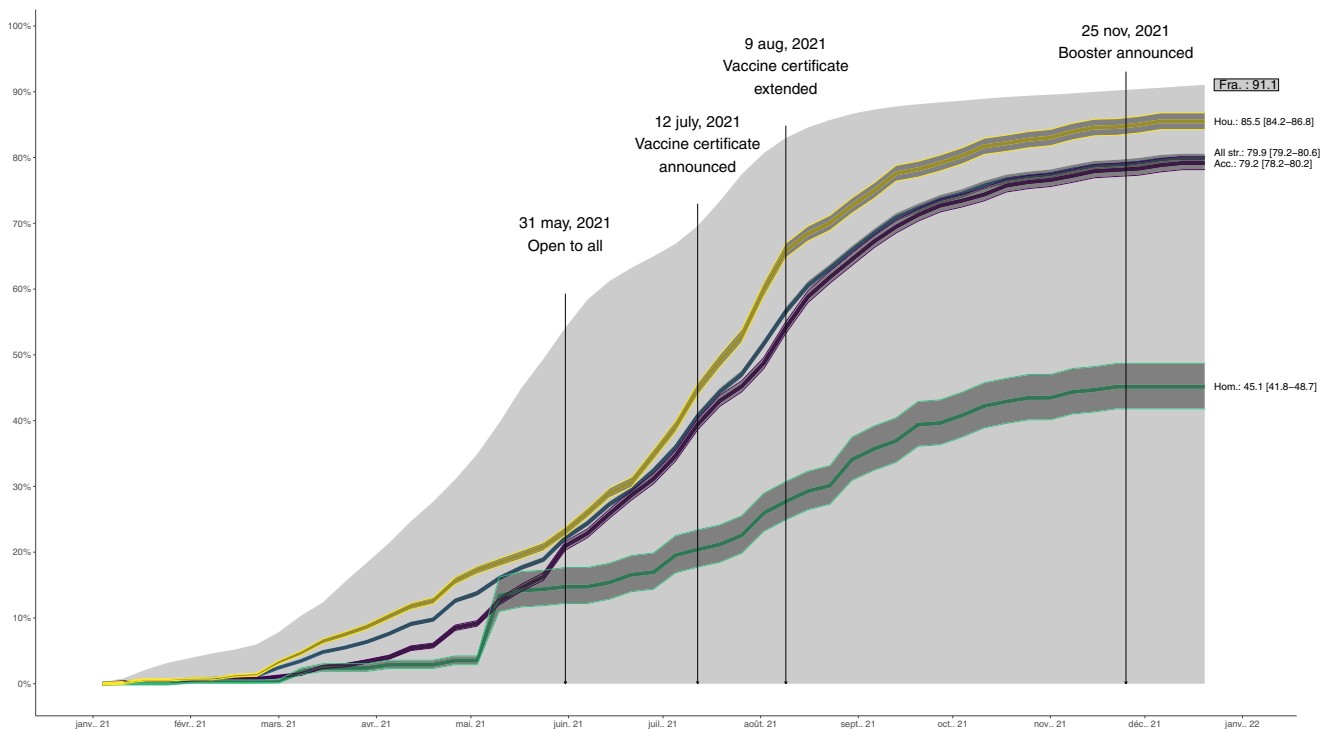

**Fig. 2 Standardized vaccination rates: strata vs French general population.** Yellow line is the standardized vaccine uptake for the Precariously Housed stratum (with corresponding 95% CIs yellow band), the purple line is the standardized vaccine uptake for Accommodated stratum and the green line is the standardized vaccine uptake for the Streets stratum. Dark blue line is the standardized vaccine uptake for all strata combined and the grey area is the standardized vaccine uptake for all French adults above 18. Sample sizes are as follows: $n = 3690$ for the Total sample representing a total standardized population of 100,567, $n = 855$ for Precariously Housed representing a standardized population of 22,788, $n = 1321$ for Accommodated representing a standardized population of 73,159 and $n = 514$ for Streets representing a standardized population of 4620. Population size for French adults above 18 was 52,751,109 as of December 28, 2021. Source data for this figure are available in Supplementary Data 10. A reproducibility guide for figures and tables is available in Supplementary Methods 3.

20.3–27.5) and more rarely outreach/onsite vaccination activities targeting PEH (18.0%; 95% CI 15.2–20.9).

Reasons for vaccination did not differ across strata. Most participants reported accepting vaccination to protect themselves (63.0%; 95% CI 60.3–65.6), as their civic duty to protect everyone (32.7%; 95% CI 29.2–36.2) or to protect vulnerable relatives (24.9%; 95% CI 21.6–28.2). Many participants reported feeling compelled to accept vaccination (44.1%; 95% CI 41.4–46.8), either to keep their job, to travel abroad, or to obtain a vaccine certificate; 23.9% (95% CI 21.3–26.5) declared the vaccine certificate as the main reason for vaccination.

Half of the non-vaccinated individuals reported having no intent to be vaccinated in the future (50.3%; 95% CI 54.7–55.0) but these comprised the majority in the Streets stratum (71.5%; 95% CI 62.3–80.7). Reasons for non-vaccination were generally linked with refusal and hesitancy (all strata: 75.6%; 95% CI 71.7–79.5), more than physical or practical barriers (all strata: 24.4, 95% CI 20.5–28.3). Precariously Housed and Accommodated strata did not differ in terms of specific reasons, with fear of immediate side effects being predominant (53.9%; 95% CI 42.5–65.3 and 59.7%; 95% CI 53.5–65.9, respectively) followed by fear of injection/fear of serious disease (43.6%; 95% CI 32.8–54.5 and 50.9%; 95% CI 44.2–57.6, respectively), and scepticism about vaccine effectiveness/utility (29.3%; 95% CI 18.5–40.1 and 25.5%; 95%CI 19.1–31.9, respectively). In the Streets stratum, participants were more subject to conspiracy theories/denial of the crisis (28.7%; 95% CI 18.5–38.9) and were more likely to be influenced by peers (31.9%; 95% CI 19.7–44.2]). All aforementioned results are summarized in Supplementary Data 4.

**Drivers of >1 dose intake**. Factors associated with vaccine uptake in univariate analysis are summarized in Supplementary Data 5. Stratum is strongly associated with vaccination ($p < 0.0001$). Sociodemographic characteristics, opinions on vaccination, need for vaccine certification, food security, support, and coping mechanisms, COVID-19 information sources, trust in authorities, health-related variables, and site-related variables were all candidates for multivariable analysis ($p$-values $< 0.001$).

Supplementary Data 6 and Fig. 4 summarize the results of the final multivariable model after backward selection. Odds of vaccine uptake vary by stratum: compared to Precariously Housed, Accommodated did not differ, but Streets individuals were less likely to be vaccinated (AOR 0.38, 95% CI 0.25–0.57, $p < 0.001$).

**Opinions on vaccination**. Participants with negative opinions about vaccines, in general, were less likely to be vaccinated (AOR 0.6, 95% CI 0.5–0.8), as were participants with negative perceptions of vaccine utility (AOR = 0.2, 95% CI 0.1–0.3) or participants afraid of the vaccine or its effects (AOR = 0.6, 95% CI 0.4–0.7).

**Sociodemographic drivers**. Odds of vaccine uptake increased with age (AOR for age 35–65 vs. 18–35; 1.3, 95% CI 1.0–1.5, AOR for >65 vs. 18–35, 2.1; 95% CI 1.3–3.7). Undocumented participants were more likely to be vaccinated than those with French or EU citizenship (AOR = 1.8, 95% CI 1.3–2.4), as were those with residence permits or refugee status (AOR = 1.7, 95% CI 1.2–2.4) and asylum seekers (AOR = 2.0, 95% CI 1.5–2.7). Vaccine uptake odds were lower for participants living with family (AOR = 0.7, 95% CI 0.5–1.0).

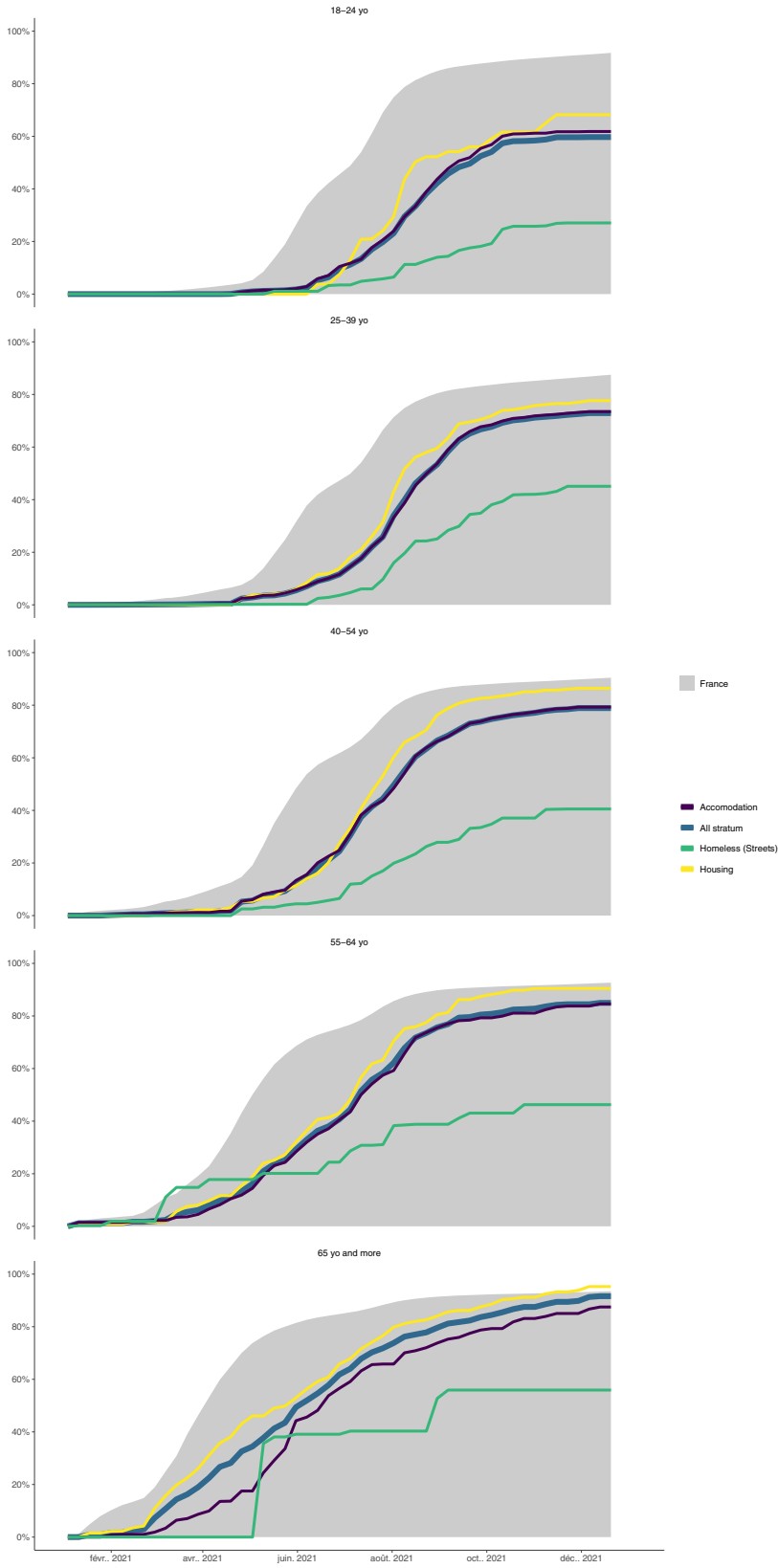

**Fig. 3 Standardized vaccination rates by age categories: strata vs French general population.** Yellow line is the standardized vaccine uptake for the Precariously Housed stratum (with corresponding 95% CIs yellow band), the purple line is standardized vaccine uptake for Accommodated stratum and the green line is the standardized vaccine uptake for the Streets stratum. Dark blue line is the standardized vaccine uptake for all strata combined and the grey area is the standardized vaccine uptake for all French adults above 18. Sample sizes are as follows: $n = 3690$ for the Total sample representing a total standardized population of 100,567, $n = 855$ for Precariously Housed representing a standardized population of 22,788, $n = 1321$ for Accommodated representing a standardized population of 73,159 and $n = 514$ for Streets representing a standardized population of 4620. Population size for French adults above 18 was 52,751,109 as of December 28, 2021. Source data for this figure are available in Supplementary Data 11. A reproducibility guide for figures and tables is available in Supplementary Methods 3.

**Economic drivers**. Odds of vaccination were higher where participants described the provision of meals by a site manager (AOR 1.5, 95% CI 1.0–2.0), for participants dependent on food distribution (AOR 1.5, 95% CI 1.1–1.9), and conversely, where participants felt independent in terms of money and food (AOR 1.4, 95% CI 1.1–1.9). Odds were lower if meals were provided by family/friends (AOR 0.8, 95% CI 0.6–1.0).

**Source of COVID-19 information**. Vaccine uptake odds were significantly higher where COVID-19 information was obtained from site managers (AOR 1.8, 95% CI 1.0–3.1). Conversely, uptake was lower where the main source of information was the internet or social media (AOR 0.7, 95% CI 0.6–0.8).

**Vaccine certificate and Health**. Vaccine uptake odds were lower for those who never needed or used a vaccine certificate (AOR 0.3, 95% CI 0.2–0.4) and in participants with no medical coverage (AOR 0.6, 95% CI 0.4–0.7). Individuals with no regular general practitioner were less likely to be vaccinated (AOR 0.7, 95% CI 0.6–0.9).

**On-site activities**. Participants living in settings with on-site vaccination activities were more likely to be vaccinated (AOR = 1.4, 95% CI 1.1–1.8).

Results of negative binomial analysis at the site level and stratified multivariable analyses are available in Supplementary Information (Supplementary Data 7–10).

## Discussion

In summary, our study highlights that PEH/PH in France are less likely to receive COVID-19 vaccination than the general population (79.9% vs. 91.1%) but also underscores that vaccine uptake varied massively according to precariousness and social integration: those living in the streets, in camps or squats, are much less likely to be vaccinated (42% with at least 1 dose) than Accommodated (75%) and Precariously Housed participants (85%). Housing is thus the most important factor linked with vaccine uptake.

Our multivariable analysis also reveals that older people, undocumented migrants and refugees, people needing the vaccine certificate, people with medical coverage, and people followed by a GP and/or social workers have higher chances of being vaccinated. Actual on-site vaccination by mobile teams or by the site manager undoubtedly increased vaccine uptake as well. On the other hand, individual factors such as negative influence by peers, negative perceptions, or a fear of the vaccine are hindering the chances of vaccination.

To our knowledge, only a handful of studies report COVID-19 vaccine uptake in PEH/PH. Of them, only one population-based study in Toronto, Canada, included homeless individuals in person and had a design similar to ours. Carried out in 62 shelters, the authors report estimates in line with those in our Accommodated and Precariously Housed strata, as 80.4% received at least one dose of vaccine and 63.6% two or more doses[20]. Other studies, another one in Ontario, Canada, three in the United States, and one from Denmark, used registries-mining to derive coverage estimates for homeless individuals[19,21–24]. Three stand out for their quality and pertinence. The first examined uptake among a convenient sample of persons experiencing homelessness in six jurisdictions of the USA and reported vaccine uptakes ranging from 22.0% to 52.0%, as compared to a range from 46.5% to 65.7% in the respective general populations[21]. In Denmark, nationwide cumulative vaccine uptakes by population groups were 54.6% in 15–64 years and 78.0% in >65 years[23]. In Ontario, 61.4% of recorded homeless

individuals received at least one dose, and 47.7% two doses[24]. Results of the three studies are comparable to those found in our Streets stratum (42%).

Apart from these studies, most of the existing evidence has assessed Covid-19 vaccine coverage in the general population in relation to income and/or ethnicity[26,27] or has focused on vaccine intention or hesitancy in PEH[28–31]. However, little evidence reports on drivers of COVID-19 uptake in the type of population we have studied, in France or elsewhere. A few studies explored factors associated with vaccine uptake in the general populations in Hong Kong, Ethiopia, and Somaliland[32–34], or in specific subgroups like very hesitant adults in the US[35], ethnic minorities in the Netherlands[36] or Palestine health workers[37].

Noticeably, some of the factors linked with vaccine uptake we report here have been detailed for other vaccine types in PEH[8–11,15] and migrants[10,13,14], and are summarized in two systematic reviews[38,39].

One of the main factors is age, with differences in vaccine uptake between older and younger adults potentially explained by a higher perception of COVID-19 risks for the elderly and by complacency for the younger ones, two well-described findings[32–37,40,41]. The introduction of vaccine certificates in July 2021 was intended to mitigate such effects and encourage younger age groups to uptake vaccination. While the impact of the certificate on vaccine uptake is obvious in our study, since this was the main reason for vaccination for 40%, it also seems to wane over time as the plateau observed in all strata and in the general population clearly shows. COVID-19 vaccine mandates are not well studied in the literature, but they were shown to be positively associated with COVID-19 vaccine uptake in the general population of Hong Kong and in the ethnic minorities in the Netherlands[32,36]. Recent studies have since questioned the impact of such policies in the long term[42,43]. Thus, differences in vaccine uptake between strata cannot be solely explained by the introduction of certificates.

Practical or physical obstacles to vaccine uptake were rare, as compared to personal motivations. Vaccine hesitancy or negative views on vaccination were the main factors associated with lower coverage. Individuals opposed to vaccination comprised a minority in our sample (12% of non-vaccinated), as compared to hesitant people, with 54% afraid of vaccine effects. Comparable findings were found in a mixed-methods study among French PEH (52% reportedly afraid of vaccine[29]) and in a multicentric survey among undocumented migrants[40]. Association of positive views on vaccination with COVID-19 vaccine uptake was highlighted in several articles[31,33,34,40,43].

Our data illustrates that vaccinated participants hold a variety of beliefs and behaviours; a majority were convinced the vaccine is useful (64%) and protects (66% vaccinated for this reason), but a non-negligible proportion holds sceptical views or felt compelled to accept vaccination; these findings confirm other work on vaccine intentions in PEH[30,31,38,39] and in the general population[36,40]. The large proportion of non-vaccinated participants without any intention to receive the vaccine in the future is also worth noting, as this observation was also made in the survey conducted in the general population of Somaliland[34].

The effects of peer pressure, reflecting the influence of friends, relatives, and others on vaccination intentions are well-described in the literature[38,39,41,42]. Our findings are consistent with others on general vaccine intention among migrants[13,14,38,39], including some data for COVID-19[30,31].

These data underline the importance of awareness-raising and sensitization by trusted third parties such as homeless organizations, social workers, mediators, and site managers, in line with qualitative studies[29,30]. Our data also show the positive impact on vaccine uptake by support mechanisms and follow-up by medical

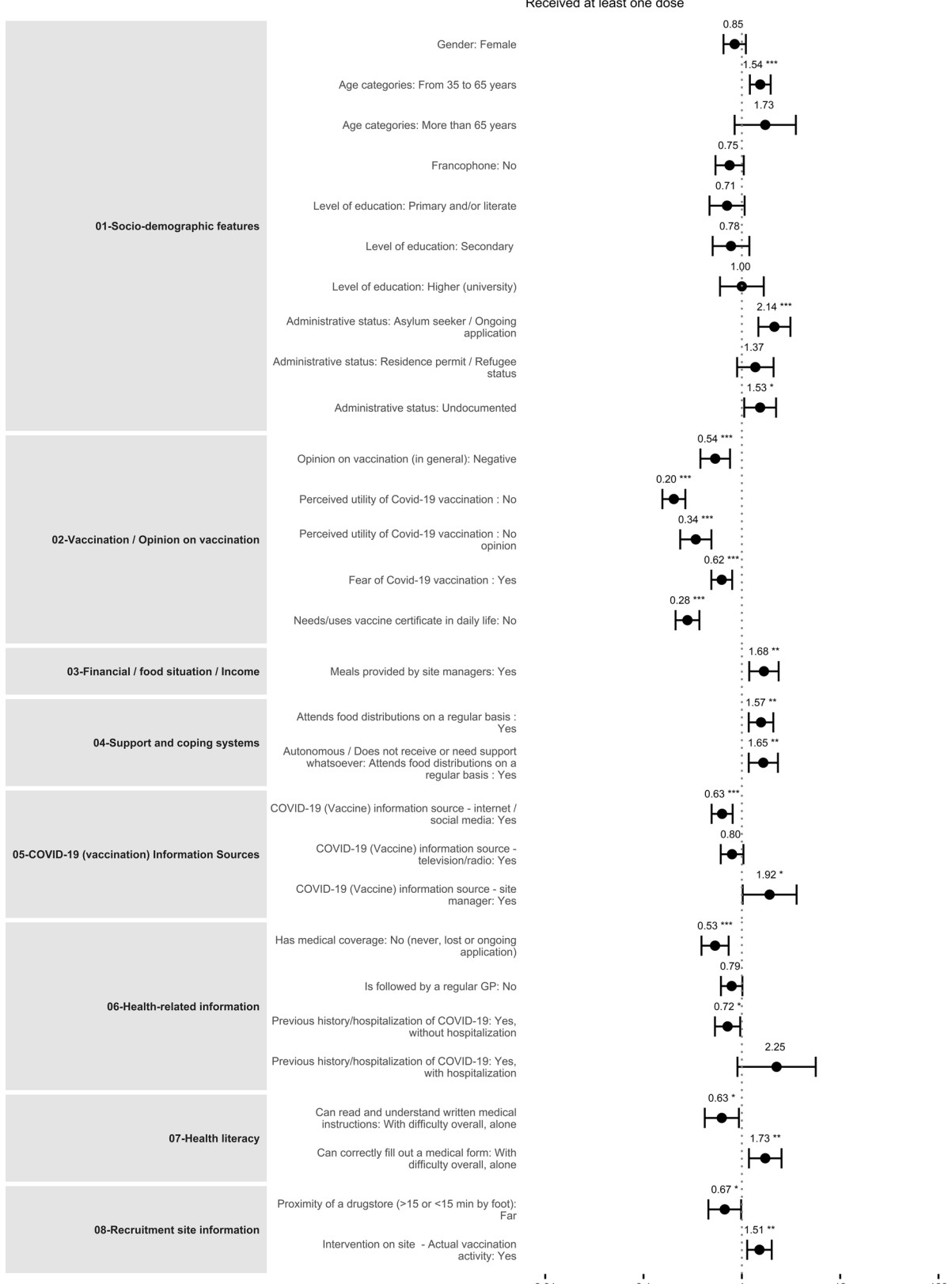

**Fig. 4 Forest plot (final multilevel mixed logistic regression).** Full dots represent estimates of Adjusted Odds-Ratio and fully capped lines their corresponding 95% Confidence Intervals. Asterisks indicate significant associations and their strength (* for *p*-values < 0.05, ** for *p*-values < 0.01 and *** for *p*-values < 0.001). Sample size for the final model was 3508. Source data are available in Supplementary Data 6. A reproducibility guide for figures and tables is available in Supplementary Methods 3.

professionals and/or by social organizations. Such factors are rarely reported in the literature[38,39,44], although some data documents the effects of lack of support from health personnel or even their negative influence on vaccination[29,38–40,45,46].

We also report on structural barriers to vaccine uptake, including distance/time to vaccination centres, lack of information on locations/dates, and problems making online appointments. The more minor role played by these factors may reflect national policies ensuring free vaccination, regardless of medical coverage or administrative status, and increased access to vaccines in the tight network of vaccination centres, drugstores, and GPs[16]. As a complement, the deployment of mobile teams and specialized site-based vaccination activities by homeless organizations and NGOs may have helped to reach those with lower access to care, as it has been the case in other campaigns[9,13,14,39] and for COVID-19[16,40].

Participants in our sample reported lacking awareness of such strategies or misjudged their rights, with many mistakenly believing they lacked entitlement. Notably, we found that those with medical cover, registered with a GP, or who had recourse to the healthcare system, were more likely to access vaccination, in line with prior work on COVID-19[39,40,44,46] and other conditions[13,38]. Misinformation due to rumours, and defiance towards authorities reinforce these barriers and are well-known in PEH[13,30,31,38,39,44,46–48].

**Strengths and limitations**. Study strengths include efforts to ensure rigorous methodology, including the use of a face-to-face survey (rare in PEH studies), conducted in eight of the most commonly used languages in this population.

The most innovative aspect is the sampling process, in which every effort is made to ensure the representativeness of all PEH/PH categories and housing types with stratification based on the ETHOS typology and random selection of sites and individuals.

One limitation is the high replacement rate, either because of absence or refusal. In the Accommodated stratum, the proportion of women was higher than expected and vaccine uptake therefore may be underestimated, since women are less vaccinated overall in our study (72.4% vs. 79.4%, $p < 0.001$; Supplementary Data 3) and are reportedly less vaccinated in general population[32,35,43] and in specific groups[36,37]. In the Precariously Housed stratum, elderly and/or retired people were also over-represented compared to people of working age, despite efforts to visit sites outside work hours. This would likely lead to an overestimation of vaccine uptake, given that elderly people are generally better vaccinated overall. Language barriers may also have contributed to increased refusals.

In addition, the survey took place at the beginning of winter, which may have influenced the results for the Streets stratum: census data used for the sampling frame dated back to March 2021 and the Plan Grand Froid (sheltering of people on the street during winter; Supplementary Notes 2) was already in place. Paradoxically, this fact warrants better representativeness in this population.

Finally, cross-sectional studies limitations may also apply, specifically the inability to adjust on non-measured confounding factors, the potential for social desirability bias in responses (relating to the support received and reasons for vaccination or non-vaccination), and survival or healthy worker bias; only those present and in good health could be interviewed.

### Conclusion

We found that COVID-19 vaccine uptake and coverage are lower for this PEH/PH population. Coverage was higher amongst those with access to the common law system and/or accompanied or supported by associations. The national vaccination strategy, tailored for the general population, seemed to also have reached Precariously Housed and Accommodated PEH/PHs as demonstrated by the high proportion of vaccinated in dedicated centres or drugstores, mostly pushed by the vaccine mandate. However, despite the involvement of social and humanitarian actors, substantial effort is still needed to reach the most excluded, street-sleeping individuals.

Our findings have implications for policy regarding these vulnerable groups. Outreach activities and on-site vaccination programs should be extended and tailored to targeted subgroups. Sensitization activities involving field actors who work closely with PEH/PH populations should take place early in such vaccination campaigns to address barriers like vaccine hesitancy and complacency. Our study reveals that high levels of vaccination can be obtained even in these vulnerable groups; higher uptake amongst the Precariously Housed and Accommodated strata implies that policies ensuring free, universal access to vaccination and the support of field actors achieve high coverage.

This work highlights the role of determinants relating to social integration and housing in relation to vaccination. The overriding influence of housing insecurity, especially, suggests that policies prioritizing secure housing first, to address health and social needs, may have value.

These insights may apply to activities and planning for future pandemics but remain valid well beyond the COVID-19 pandemic, when providing operational support and care to vulnerable populations, for example.

Our study cannot easily be extrapolated to other contexts. Nonetheless, our lessons could be of use in countries with a similar migration and/or homelessness profile or in countries with homelessness support policies similar to French ones. In any case, more data on COVID-19 vaccine uptake and its drivers, especially from other European countries, are still deeply needed.

### Data availability

The datasets generated during and/or analysed for the current article, alongside the R scripts used for analysis, are available at https://osf.io/gpkum/. Readers who wish to use the data for commercial or non-commercial purposes should first contact the corresponding author T.R. The study protocol, detailed statistical analysis plan and electronic forms are available upon request from the corresponding author T.R.

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

## Acknowledgements

The authors want to thank all the participants in the survey, as well as the managers of the hostels and emergency shelters who made the survey possible. Authors would also like to thank partner organizations for sharing their data and allowing us to constitute sampling frames: RATP, Ville de Paris (UASA), DIHAL, SAMU Social de Paris, France Terre d'Asile, Emmaüs Solidarité and Prospective & Coopération. The authors want to thank Jalpa Shah, Cécile Allaire and Elodie Richard for their insight during protocol and questionnaire development. We acknowledge the support of Emma Veitch, medical editor for MSF, in providing editorial assistance; her work was funded by MSF-USA. The study was funded by Santé Publique France, Agence Nationale de Recherches sur le Sida (ANRS-MiE/Capnet) and Agence Régionale de Santé—Ile de France with additional support provided by Médecins Sans Frontières and Société de Pathologie Infectieuse de Langue Française. External donors had no role in the study design, data collection, interpretation, analysis, report writing, or the decision to submit for publication.

## Authors contributions

T.R., B.M. and S.V. conceived the study (literature search, study design, etc.). T.R., B.M., J.S.-H., M.M. and S.V. developed the study protocol. T.R., B.M., J.S.-H., M.M. and C.V. performed field data collection (and data questionnaires) and supervised the field study. T.R. and G.L. performed data management and statistical data analysis. T.R., B.M. and

S.V. performed a literature search for the manuscript. G.L. and T.R. verified the underlying data and performed additional analyses. All authors interpreted the results, contributed to writing the manuscript, and approved the final version for submission.

**Competing interests**
The authors declare no competing interests.

**Additional information**

