## [Peer Review File · Communications Medicine]

This manuscript has been previously reviewed at another Nature Portfolio journal. This document only contains reviewer comments and rebuttal letters for versions considered at Communications Medicine

REVIEWERS' COMMENTS:

Reviewer #3 (Remarks to the Author):

I thank the authors for this revised paper. I find it has addressed most of my initial comments, and reads well.

My only remaining concern, reiterated from my previous review, regards the continued labelling of the group 'Housed', on the grounds that it may cause misinterpretation by readers. Though indeed technically housed (although in suboptimal conditions), 95% of this group are staying in Migrant Workers Hostels which isn't the kind of housing implied by the generic label 'Housed' (which brings to mind a sort of general population housed control). The most simple solution might be to relabel this group without change to the group itself (something like 'vulnerably housed', or 'precariously housed' perhaps?).

Reviewer #4 (Remarks to the Author):

(no comments to the authors were provided)

Reviewer #3 (Remarks to the Author):

I thank the authors for this revised paper. I find it has addressed most of my initial comments, and reads well.

My only remaining concern, reiterated from my previous review, regards the continued labelling of the group 'Housed', on the grounds that it may cause misinterpretation by readers. Though indeed technically housed (although in suboptimal conditions), 95% of this group are staying in Migrant Workers Hostels which isn't the kind of housing implied by the generic label 'Housed' (which brings to mind a sort of general population housed control). The most simple solution might be to relabel this group without change to the group itself (something like 'vulnerably housed', or 'precariously housed' perhaps?).

Response

We apologize to the Reviewer 3, we did fully understand his point in the previous Comments. We edited the whole manuscript and tables to reflect his fair suggestion.